# Preparation and Application of a NaCl-KCl-CsCl-Cs_2_ZrCl_6_ Composite Electrolyte

**DOI:** 10.3390/ma16062270

**Published:** 2023-03-11

**Authors:** Wenzhen Zou, Yanke Wu, Lijun Wang, Guoqing Yan, Zhaohui Ma, Jiandong Zhang

**Affiliations:** 1National Engineering Research Center for Environment-Friendly Metallurgy in Producing Premium Non-Ferrous Metals, China GRINM Group Corp., Ltd., Beijing 100088, China; 2GRINM Resources and Environment Tech. Co., Ltd., Beijing 100088, China; 3General Research Institute for Nonferrous Metal, Beijing 100088, China; 4Beijing Engineering Research Center of Strategic Nonferrous Metals Green Manufacturing Technology, Beijing 100088, China

**Keywords:** Cs_2_ZrCl_6_, preparation, molten salt electrolyte, electrolytic refining, zirconium

## Abstract

Using ternary molten salt with a molar ratio of NaCl:KCl:CsCl = 30:24.5:45.5 and ZrCl_4_ as raw materials to prepare a NaCl-KCl-CsCl-Cs_2_ZrCl_6_ composite electrolyte. Characterizing by XRD, ICP-AES, optical microscopy and SEM-EDS, the results showed that when the molar ratio of CsCl:ZrCl_4_ ≥ 2:1, Cs_2_ZrCl_6_ was generated according to the stoichiometric reaction; when the molar ratio of CsCl:ZrCl_4_ < 2:1, CsCl in molten salt was almost completely converted to Cs_2_ZrCl_6_, and there was a ZrCl_4_ phase. When the molar ratio of CsCl:ZrCl_4_ = 2:1, with the increase of the reaction temperature and reaction time, the concentration of zirconium ions first increased and then decreased. The optimized preparation process conditions are: the 2:1 molar ratio of CsCl to ZrCl_4_ in NaCl-KCl-CsCl, 500 °C of reaction temperature of and 3 h of reaction time. Under this condition, 99.68% conversion rate from ZrCl_4_ to Cs_2_ZrCl_6_ was obtained. Taking the prepared NaCl-KCl-CsCl-Cs_2_ZrCl_6_ composite electrolyte as a raw material, a preliminary study of molten salt electrolytic refining zirconium was carried out, and a refined zirconium product with a dendrite of 10.61 mm was obtained under the conditions of a zirconium ions concentration of 5%, an electrolysis temperature of 750 °C, a current density of 0.1 A/cm^2^, and an electrolysis time of 9 h, indicating that the composite electrolyte can be used for the electrolytic refining of zirconium.

## 1. Introduction

Zirconium and hafnium have excellent corrosion resistance, mechanical properties, and nuclear properties, and they are important structural and functional materials for nuclear reactors [1,2,3]. The preparation methods of zirconium and hafnium mainly include the Kroll method, the molten salt electrolysis method, and the metal thermal reduction method [4,5,6,7]. The purification methods include the molten salt electrolytic refining method, the iodization method, and the electron beam melting method [8,9].

Molten salt electrorefining is one of the main methods to purify zirconium metal. The molten salt electrolyte system mainly includes a fluoride system, a fluoride–chloride system, and a chloride system [8]. In the fluoride system, the morphological feature of the deposited Zr is found to be powdery by electrolytic refining processes such as LiF-NaF-ZrF_4_ [9,10] and LiF-KF-ZrF_4_ [11]. In the fluoride–chloride system, powdery Zr deposits are known to be produced in LiCl-KCl-LiF-ZrCl_4_ [12,13,14]. Furthermore, Chen et al. [15] and Wu et al. [16] obtained powdery and dendritic products by using K_2_ZrF_6_ as a zirconium source in NaCl-KCl-K_2_ZrF_6_. Relevant studies have shown that powdery products are easy to obtain by electrolytic refining processes in fluoride system and fluoride–chloride system, resulting in a high salt inclusion rate of cathode products and high oxygen content after subsequent water washing treatment. However, dendritic zirconium products are easier to obtain by using chloride as a molten salt electrolyte with ZrCl_4_ as a zirconium source for electrolytic refining. However, ZrCl_4_ is difficult to add to molten salt, and ZrCl_4_ volatilization loss is prone to occur during electrorefining processes. For example, Inman et al. [17], Boboian et al. [18], Sakamura et al. [19], Ghosh et al. [20], and Lee et al. [21] found ZrCl_4_ loss in LiCl-KCl-ZrCl_4_. Therefore, the preparation of a chloride electrolyte containing zirconium ions with high-temperature stability is the key to solving the problem.

In order to solve the above problems of chloride molten salt, Cai et al. [22] obtained a LiCl-KCl-ZrCl_4_ molten salt system which meets the requirements of zirconium electrolytic refining by using LiCl-KCl-CuCl as a raw material and replacing CuCl in molten salt with Zr. However, this method is still in the experimental research stage and its operation is complicated. Liu Xu et al. [23] proposed to use potassium chloride and hafnium tetrachloride as raw materials to directly prepare potassium hafnium chloride through a gas–solid reaction and the use of potassium chlorohafnium acid as the hafnium source to prepare a molten salt electrolyte for electrolysis and refining metal hafnium. However, due to the gas–solid phase reaction, the reaction rate is low, resulting in a low concentration of hafnium ions in the electrolyte, and the control of the molten salt preparation process is relatively complicated. The main reason why it is difficult to prepare ZrCl_4_/HfCl_4_ to chloride molten salts is that their sublimation temperatures are low, 331 °C and 320 °C [24], respectively, and the melting points of commonly used binary molten salts LiCl-NaCl, NaCl-KCl], NaCl-BaCl_2_, and KCl-BaCl_2_ are 553 °C [25], 657 °C [26], 648 °C, and 655 °C [27], respectively. It can be seen that the sublimation temperature of ZrCl_4_ and HfCl_4_ is lower than the melting point of the above binary molten salt.

Ternary eutectic molten salt has a lower melting point. Low melting point ternary chloride eutectic molten salt has been used to prepare metal lithium, uranium, iridium, etc. [28,29,30]. It is expected to prepare chloride composite electrolytes with high zirconium content by using ternary molten salt. Furthermore, a titanium chloride composite electrolyte was prepared by introducing low-valent titanium ions into NaCl-KCl blank salt by the reaction of tetravalent titanium and sponge titanium [31,32]. A NaCl-KCl-K_2_HfCl_6_ composite electrolyte was directly prepared by using hafnium tetrachloride and an equal molar ratio of sodium chloride and potassium chloride as raw materials [23].

The use of a ternary chloride composite electrolyte for molten salt electrolytic refining of zirconium has not been reported. Kipouros et al. [33] found that the alkali metals hexachlorozirconate and hexachlorohafnium ester are stable compounds, and the thermodynamic stability increases in the order of Li_2_MC1_6_-Na_2_MCl_6_-K_2_MCl_6_-Cs_2_MCl_6_ (M stands for Zr or Hf), that is, as the size of the alkali metal cation increases, the stability of hexachlorozirconium/hexachlorohafnium ester increases, and the activity/volatility of zirconium tetrachloride/hafnium decreases. The LiCl-KCl-CsCl system has a lower melting point, but LiCl is more absorbent and has a higher cost [27].

Therefore, in this paper, a NaCl-KCl-CsCl ternary molten salt system was used to carry out research. The chloride system can not only obtain dendritic zirconium products, but which can effectively avoid the high oxygen content caused by the powder zirconium products brought by the fluoride system and the fluoride chloride system. It can also solve the problem that the high-temperature stability of zirconium ions in chloride systems is not strong and the concentration is not high. The composite electrolyte was prepared by reacting the molten salt system with zirconium tetrachloride, and the preparation process was studied and analyzed. The obtained composite electrolyte was used to verify the electrolytic refining of zirconium.

## 2. Materials and Methods

### 2.1. Experimental Materials

The main raw materials used in the experiment are shown in Table 1.

### 2.2. Preparation Method of the Composite Electrolyte

The molar ratio of NaCl:KCl = 1:1 was weighed, mixed, and dehydrated at 300 °C for 24 h in a vacuum oven. It was then cooled after heating and melting. NaCl-KCl-CsCl molten salt was prepared by the same method. The molar ratio of NaCl:KCl:CsCl was 30:24.5:45.5. CsCl was dried after dehydration without other treatment. The obtained NaCl-KCl, CsCl, and NaCl-KCl-CsCl molten salts were stored in a glove box for use.

### 2.3. Characterization Methods of Materials

The phase composition of the composite electrolyte was analyzed by X-ray diffraction (XRD). The concentration of zirconium ions in the composite electrolyte was analyzed by an inductively coupled plasma emission spectrometer (ICP-AES). The microstructure, morphology, and elemental analysis of the electrolytic refined zirconium dendrites were characterized by an optical microscope, a scanning electron microscope (SEM), and an X-ray energy dispersive spectrometer (EDS).

## 3. Results and Discussion

Firstly, the feasibility of preparing a low melting point chloride composite electrolyte by a NaCl-KCl-CsCl ternary molten salt system was verified. Then, factors such as the type and ratio of raw salt, the reaction temperature, and the reaction time were studied experimentally. The overall experimental scheme is shown in Table 2.

### 3.1. Feasibility of Adding ZrCl_4_ in Different Molten Salts

NaCl-KCl, CsCl, NaCl-KCl-CsCl, and ZrCl_4_ were fully mixed according to the experimental numbers in Table 2 and reacted at 550 °C for 3 h. The XRD analysis of the product is shown in Figure 1.

It can be seen from Figure 1 that only NaCl-KCl-CsCl ternary molten salt and ZrCl_4_ can prepare Cs_2_ZrCl_6_ at 550 °C. There is almost no reaction between the pure CsCl system and ZrCl_4_, and the product is mainly CsCl. No ZrCl_4_ phase and Cs_2_ZrCl_6_ phase are found. This is because the reaction temperature is much higher than the sublimation point of ZrCl_4_, resulting in the complete volatilization of ZrCl_4_. There is almost no reaction between the NaCl-KCl system and ZrCl_4_. The products are mainly NaCl and KCl, and no K_2_ZrCl_6_ and ZrCl_4_ phases are found. The ZrCl_4_ phase was not found in the reaction product of NaCl-KCl-CsCl with ZrCl_4_. The phase detected in the product was mainly Cs_2_ZrCl_6_, and also included NaCl, KCl, and CsCl, indicating that the low melting point of NaCl-KCl-CsCl containing CsCl was beneficial to the addition of ZrCl_4_ and CsCl had a strong interaction with ZrCl_4_ in the molten salt. It is feasible to prepare molten salt containing zirconium ions by a NaCl-KCl-CsCl ternary molten salt system, and zirconium ions have good stability in molten salt. In the following section, the effects of process conditions on the preparation of composite electrolytes were studied for the NaCl-KCl-CsCl ternary system.

### 3.2. Effect of Raw Material Ratio on the Preparation of Composite Electrolyte

The raw material ratio is one of the important factors in the preparation of compounds. The NaCl-KCl-CsCl ternary molten salts with CsCl:ZrCl_4_ molar ratios of 4:1, 2:1, and 1:1 were fully mixed with ZrCl_4_, respectively, and reacted at 550 °C for 3 h. The experimental conditions were numbered 4, 6, and 5 in Table 2, and the XRD analysis of the product is shown in Figure 2.

The results of Figure 2 show that there are some differences in the phase of the product after the reaction under different raw material ratios, but the main phase of the product is Cs_2_ZrCl_6_. For the experimental No. 4 of CsCl excess, the main substances after reaction were CsCl, Cs_2_ZrCl_6_, and a small amount of NaCl, KCl, no ZrCl_4_ phase, and the CsCl peak was the strongest, indicating that ZrCl_4_ completely combined with CsCl to form Cs_2_ZrCl_6_, and excess CsCl existed independently. When the molar ratio of CsCl to ZrCl_4_ is 2:1, that is, according to experiment No.6 of the stoichiometric ratio of Cs_2_ZrCl_6_, the product phases include NaCl, KCl, CsCl, and Cs_2_ZrCl_6_. There is no ZrCl_4_ phase, there is a weak CsCl diffraction peak, and Cs_2_ZrCl_6_ is the main phase, indicating that the ZrCl_4_ in the raw material almost completely reacts with CsCl and transforms into Cs_2_ZrCl_6_. When the molar ratio of CsCl to ZrCl_4_ is 1:1, that is, the amount of ZrCl_4_ is twice as much as the stoichiometric ratio of Cs_2_ZrCl_6_, the product phase includes NaCl, KCl, CsCl, and Cs_2_ZrCl_6_ and ZrCl_4_ five phases, the CsCl diffraction peak is weak, and Cs_2_ZrCl_6_ is the main phase. There is a ZrCl_4_ phase in the reaction product because the amount of ZrCl_4_ added exceeds the equivalent of Cs_2_ZrCl_6_.

The analysis results of zirconium ion concentrations in the three kinds of molten salts are listed in Table 3. Compared with the addition amount, when the ratio of NaCl-KCl-CsCl to ZrCl_4_ is 1:1 and 1:2, the zirconium ion concentration in the molten salt is basically the same as the additional amount. When the ratio is 1:4, the zirconium ion concentration in the molten salt is slightly lower than the additional amount, indicating that there is a certain volatilization of ZrCl_4_, which is consistent with the XRD analysis results.

Considering the factors such as costs and the volatility of ZrCl_4_, it is suggested that the ratio of NaCl-KCl-CsCl to ZrCl_4_ should be 2:1.

### 3.3. Effect of Reaction Temperature on the Preparation of Composite Electrolyte

Under the condition that the molar ratio of NaCl-KCl-CsCl to ZrCl_4_ is 2:1 and the reaction time is 3 h, the effects of phase difference on the preparation of composite electrolyte were studied at 450 °C, 500 °C, 550 °C, 600 °C, and 800 °C. The experimental conditions are as follows: Table 2, the experimental numbers 7, 6, 8, 9, and 10. The XRD analysis results of the products under different temperature conditions are shown in Figure 3, and the products contain NaCl, KCl, CsCl, and Cs_2_ZrCl_6_ as four phases. The main difference between the product phases at different temperatures is that the peak intensity of the Cs_2_ZrCl_6_ phase and the CsCl phase is obviously different. With the increase in temperature, the diffraction peak intensity of Cs_2_ZrCl_6_ increases first and then decreases. This is due to the fact that the reaction rate is small and the reaction is slow when the temperature is low, and there is still unreacted CsCl after 3 h of reaction. Until the temperature reaches 500 °C, the diffraction peak intensity of Cs_2_ZrCl_6_ reaches the strongest; as the temperature further increases, CsCl gradually replaces Cs_2_ZrCl_6_ as the main phase of the post-reaction material, which is due to the increased volatilization of ZrCl_4_ at high temperatures, resulting in a gradual decrease in the production of Cs_2_ZrCl_6_.

The analysis results of zirconium ion concentration in the composite electrolyte under different temperature conditions are shown in Figure 4. It can be seen that with the increase in temperature, the concentration of Zr ions shows a consistent trend with the results of the XRD, which indicates that 500 °C is the better reaction temperature.

Too low or too high a temperature is not conducive to the formation of Cs_2_ZrCl_6_ in the reaction of molten salt. Based on the above analysis, the suitable temperature for preparing high zirconium ion concentration molten salt is 500 °C.

### 3.4. Effect of Reaction Time on the Preparation of the Composite Electrolyte

Under the conditions of the NaCl-KCl-CsCl and ZrCl_4_ molar ratio, the CsCl:ZrCl_4_ molar ratio of 2:1, and the reaction temperature of 500 °C, the effects of reaction time of 2, 3, 4, 5, and 6 h on the concentration of zirconium ions in the product were studied. The experimental conditions are as follows: experimental numbers 11, 8, 12, 13, and 14 in Table 2. The analysis results of zirconium ion concentration in the product are shown in Figure 5. With the increase in reaction time, the concentration of zirconium ion increased first and then decreased gradually. When the reaction time was 3 h, the highest concentration was 12.50%, which was the closest to the concentration of 12.54%.

### 3.5. Preparation of the NaCl-KCl-CsCl-Cs_2_ZrCl_6_ Composite Electrolyte

According to the above experiments, the optimized process for the preparation of the Cs_2_ZrCl_6_-containing composite molten salt was determined: the molar ratio of CsCl to ZrCl_4_ in the ternary molten salt was controlled at 2:1, the reaction temperature was 500 °C, and the reaction time was 3 h. The composite electrolyte was prepared. According to the solid salt raw material and the main phase, it was named as a NaCl-KCl-CsCl-Cs_2_ZrCl_6_ composite electrolyte. The appearance of the solidified composite electrolyte is shown in Figure 6, and the XRD analysis results are shown in Figure 7. The main phase is Cs_2_ZrCl_6_. The results of zirconium ion concentration in the composite electrolyte showed that the conversion of ZrCl_4_ to Cs_2_ZrCl_6_ was 99.68%.

### 3.6. Application of the NaCl-KCl-CsCl-Cs_2_ZrCl_6_ Composite Electrolyte

To verify its performance, the NaCl-KCl-CsCl-Cs_2_ZrCl_6_ composite electrolyte was mixed with the NaCl-KCl (NaCl:KCl = 1:1) binary electrolyte to prepare a mixed salt containing 3% zirconium ion concentration. A stainless steel rod was used as the cathode, a graphite crucible was used as the anode, and sponge zirconium was used as the anode material. The electrolytic refining experiment was carried out at a current density of 0.1 A/cm^2^, an electrolysis temperature of 750 °C, and an electrolysis time of 9 h. The obtained electrolytic refined zirconium cathode deposition product is shown in Figure 8, which is obviously dendritic.

The cathodic deposition product was dried after pickling and washing to obtain electrolytically refined zirconium dendrites. Observed under an optical microscope, as shown in Figure 9, it is obviously dendritic. Figure 9b is the result of the high magnification observation highlighted in Figure 9a, showing that the size of the product is large, up to 10.61 mm.

SEM-EDS analysis of the small-sized electrolytic refined zirconium dendrites is shown in Figure 10, which further shows that the obtained products have obvious dendrites.

## 4. Conclusions

In this paper, the effects of the molten salt system, raw material ratio, reaction temperature, and reaction time on the preparation of a composite electrolyte were analyzed by controlling the experimental conditions. After optimizing the process conditions, the composite electrolyte was prepared and applied to the experiment of molten salt electrolytically refined zirconium. The reaction products were systematically characterized by relevant test methods. The results show that:

(1) CsCl preferentially reacts with ZrCl_4_ to form stable Cs_2_ZrCl_6_ in a NaCl-KCl-CsCl-ZrCl_4_ molten salt system, and CsCl plays a key role in the stability of zirconium ions.

(2) It was found that the optimum process conditions for preparing the NaCl-KCl-CsCl-Cs_2_ZrCl_6_ composite electrolyte were as follows: a NaCl-KCl-CsCl ternary molten salt system was used to eutectic with ZrCl_4_, the molar ratio of CsCl to ZrCl_4_ in NaCl-KCl-CsCl was controlled at 2:1, the reaction temperature was 500 °C, and the reaction time was 3 h. Under the optimized process conditions, the conversion rate of ZrCl_4_ to Cs_2_ZrCl_6_ was 99.68%.

(3) The molten salt electrolytic refining of zirconium was carried out using the prepared NaCl-KCl-CsCl-Cs_2_ZrCl_6_ composite electrolyte as a raw material. Zirconium products with a dendritic size of 10.61 mm were obtained, indicating that the composite electrolyte is suitable for zirconium electrolytic refining.

## Figures and Tables

**Figure 1 materials-16-02270-f001:**
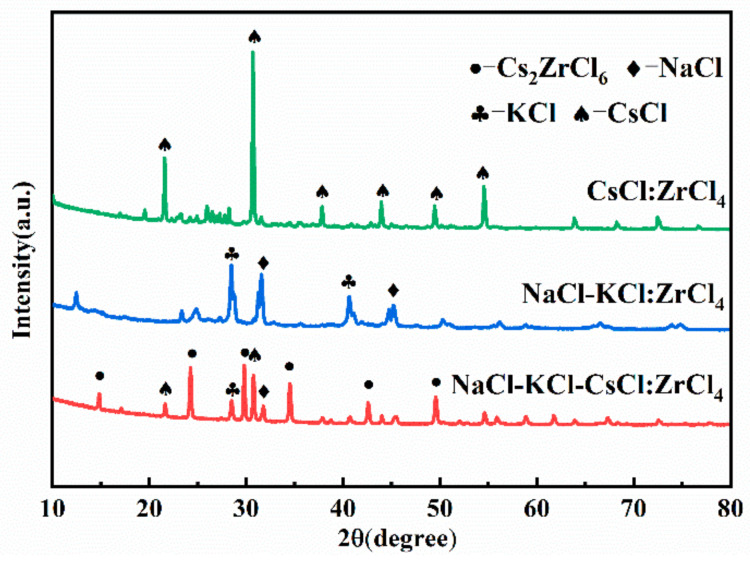
XRD pattern of products under different molten salt systems.

**Figure 2 materials-16-02270-f002:**
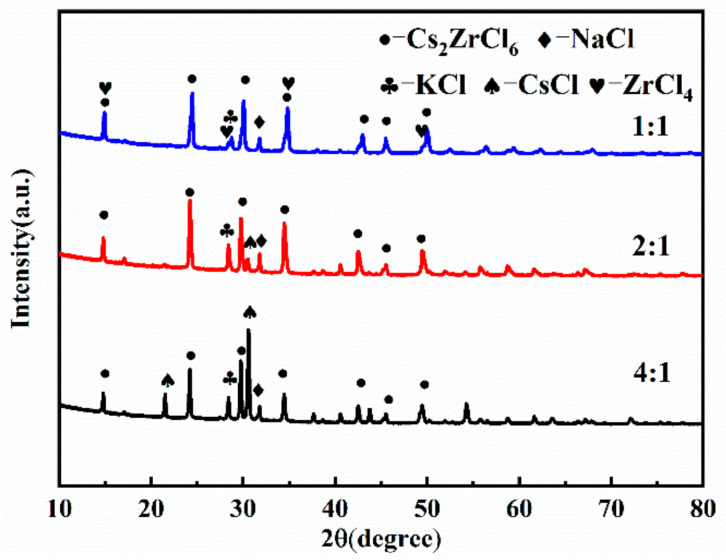
XRD pattern of products under different raw material ratio conditions.

**Figure 3 materials-16-02270-f003:**
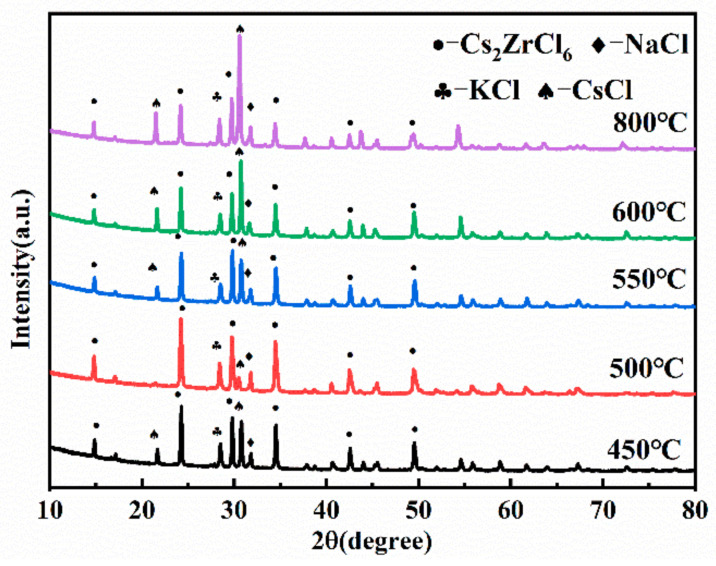
XRD pattern of products under different temperatures.

**Figure 4 materials-16-02270-f004:**
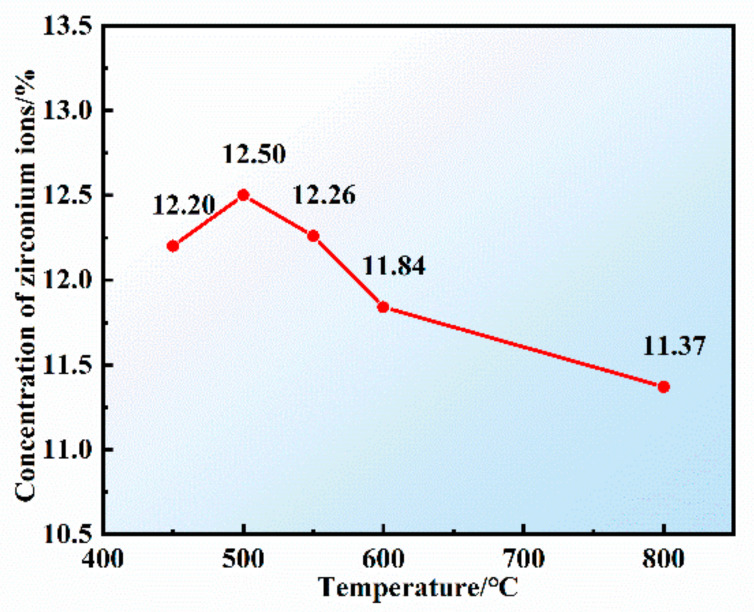
Changes in the concentration of zirconium ions in the product under different temperature conditions.

**Figure 5 materials-16-02270-f005:**
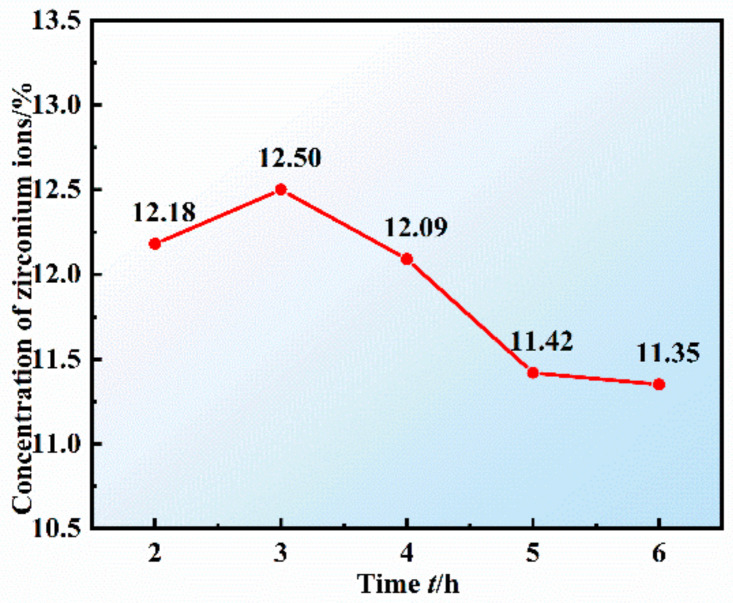
Change in zirconium ions concentration in the product under different reaction time conditions.

**Figure 6 materials-16-02270-f006:**
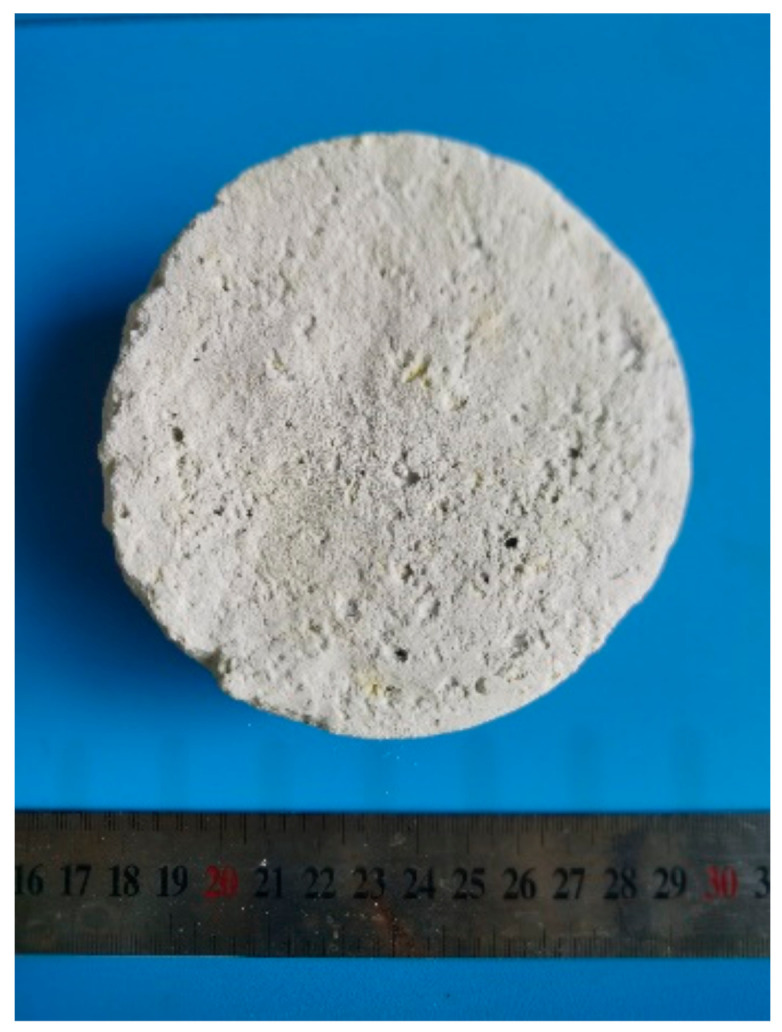
NaCl-KCl-CsCl-Cs_2_ZrCl_6_ composite electrolyte.

**Figure 7 materials-16-02270-f007:**
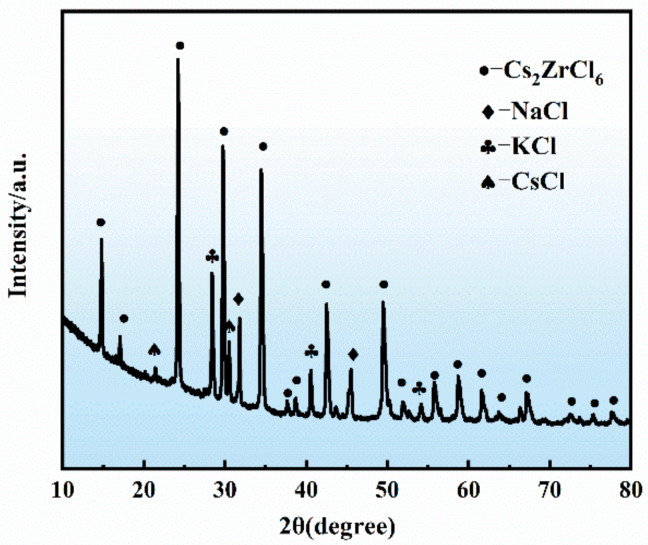
XRD patterns of the NaCl-KCl-CsCl-Cs_2_ZrCl_6_ composite electrolyte.

**Figure 8 materials-16-02270-f008:**
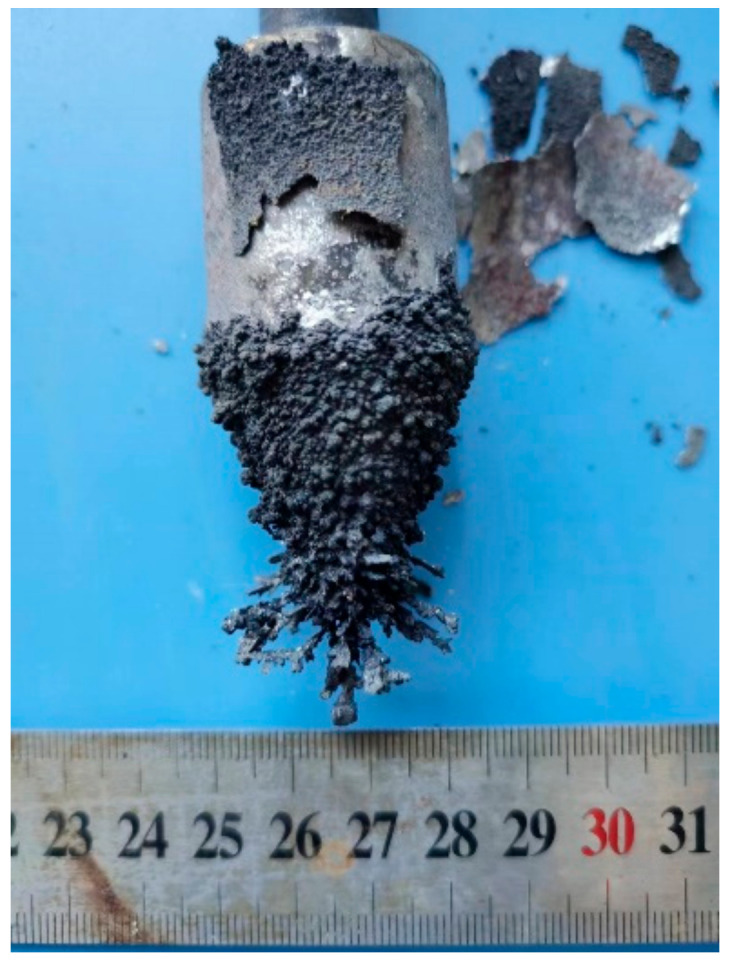
Zirconium cathode deposition products by electrolytic refining.

**Figure 9 materials-16-02270-f009:**
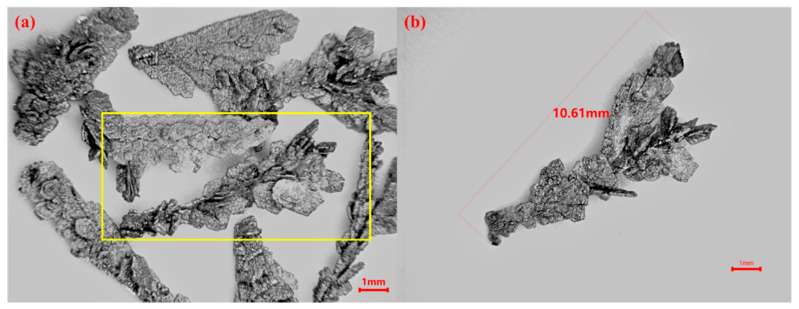
Electrolytically refined zirconium dendrites under optical microscopy. (**a**) low magnification (**b**) high magnification of the highlighted in Figure 9a.

**Figure 10 materials-16-02270-f010:**
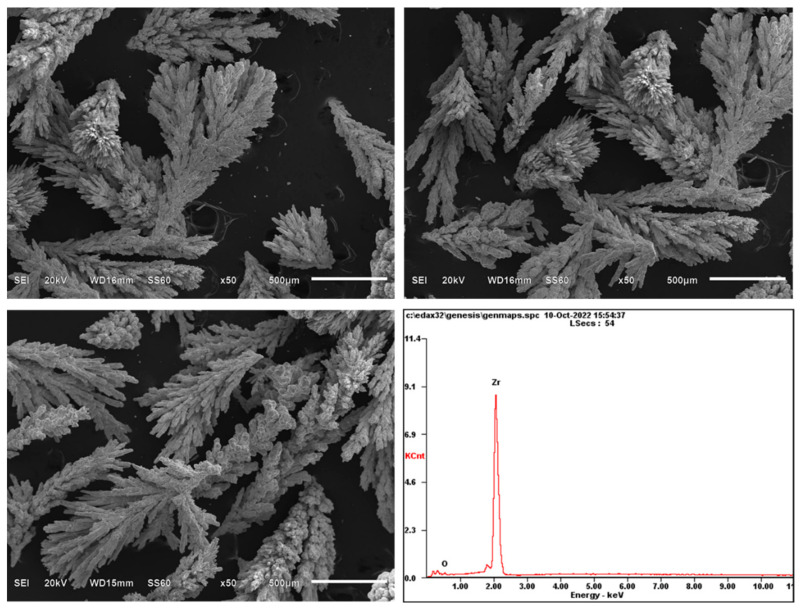
SEM and EDS analysis of electrolytically refined zirconium dendrites.

**Table 1 materials-16-02270-t001:** Main experimental raw materials.

Raw Material	Purity	Source
NaCl	Analytical reagent	Guangxi Xilong Chemical Co., Ltd.
KCl	Analytical reagent	Guangxi Xilong Chemical Co., Ltd.
CsCl	Analytical reagent	Guangxi Xilong Chemical Co., Ltd.
ZrCl_4_	Analytical reagent	Guangdong Oriental Zirconium Co., Ltd.
Zirconium sponge	Industry grade	Guangdong Oriental Zirconium Co., Ltd.

**Table 2 materials-16-02270-t002:** Design of experiments.

No.	Molten Salt System	Ratio	Temperature	Time
1	NaCl-KCl:ZrCl_4_	2:1	550 °C	3 h
2	CsCl:ZrCl_4_	2:1	550 °C	3 h
3	NaCl-KCl-CsCl:ZrCl_4_	2:1	550 °C	3 h
4	NaCl-KCl-CsCl:ZrCl_4_	4:1	550 °C	3 h
5	NaCl-KCl-CsCl:ZrCl_4_	1:1	550 °C	3 h
6	NaCl-KCl-CsCl:ZrCl_4_	2:1	550 °C	3 h
7	NaCl-KCl-CsCl:ZrCl_4_	2:1	450 °C	3 h
8	NaCl-KCl-CsCl:ZrCl_4_	2:1	500 °C	3 h
9	NaCl-KCl-CsCl:ZrCl_4_	2:1	600 °C	3 h
10	NaCl-KCl-CsCl:ZrCl_4_	2:1	800 °C	3 h
11	NaCl-KCl-CsCl:ZrCl_4_	2:1	500 °C	2 h
12	NaCl-KCl-CsCl:ZrCl_4_	2:1	500 °C	4 h
13	NaCl-KCl-CsCl:ZrCl_4_	2:1	500 °C	5 h
14	NaCl-KCl-CsCl:ZrCl_4_	2:1	500 °C	6 h

**Table 3 materials-16-02270-t003:** Product zirconium ions concentration under different raw material ratio conditions.

No.	CsCl:ZrCl_4_	Added Concentration (%)	Post-Reaction Concentration (%)
4	4:1	7.47	7.32
6	2:1	12.54	12.50
5	1:1	19.00	17.72

## Data Availability

Not applicable.

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
