# Peer review of "Preparation and Application of a NaCl-KCl-CsCl-Cs2ZrCl6 Composite Electrolyte"

_materials, 2023, doi:10.3390/ma16062270_

Round 1

Reviewer 1 Report

Having examined your manuscript entitledPreparation and application of NaCl-KCl-CsCl-Cs2ZrCl6 composite electrolyte”, I note that you have optimized and made use of the molten salts route  to obtain NaCl-KCl-CsCl-Cs2ZrCl6 composite electrolyte but before their publication some issues should be addressed. Therefore, it requires a major revision. Please check carefully. The questions and comments for this manuscript are as follows:

(1)    Results

Please define why you have selected as first instance the use of a ratio 2:1.

Please include in the bottom of the XRD results the pdf cards of each component, to follow which phases you have.

How you determine the concentration of zirconium ions in figure 4, please explain the accuracy of the experiments. By XRD you probably can determine the concentration of Cs2ZrCl6, the trend is similar that the one observed by ICP-AES. Please revise same issue in Figure 5.

Please explain the selection of the parameters in the  electrolytic refining experiment.

Correct the numeration of figures, figure 8 and 6.

Please explain the advantages of obtained zirconium dendrites and tehri role I the electrolyte.

(2)    Conclusions

Please explain the novelty of the review It is not clear which dendrites sizes are required and have already used.

Please revise the statement the maximum grain 264 size could reach 2.3 mm. It is not possible to observe grain in the SEM that you presented. Take in account the concept of particle size, grain size and crystallite size.

Author Response

Response to Reviewer 1 Comments

Dear reviewer:

Re: Manuscript ID: materials-2216358 and Title: Preparation and application of NaCl-KCl-CsCl-Cs2ZrCl6 composite electrolyte.

Thank you from the bottom of my heart for your recognition and comments on our manuscript. Those comments are valuable and helpful. We have read through comments carefully and have made corrections. Based on your valuable comments, we have uploaded the revised manuscript file. Revisions in the text are shown using red highlight for additions, and strikethrough font for deletions. The responses to the your comments are marked in red and presented following.

We would love to thank you for allowing us to resubmit a revised copy of the manuscript and we highly appreciate your time and consideration.

Sincerely.

WenZhen Zou.

(1)    Results

Point 1: Please define why you have selected as first instance the use of a ratio 2:1.

Response 1: Thanks for your comments.

2CsCl(s)+ZrCl4(g)→Cs2ZrCl6(s)

The ratio 2 : 1 is used as the first example because of the stoichiometric ratio Cs : Zr = 2 : 1.

Point 2: Please include in the bottom of the XRD results the pdf cards of each component, to follow which phases you have.

Response 2: Thanks for your comments. Inserting the pdf card of each component will make it difficult to control the size of some XRD results. Therefore, I analyzed and marked each component through Jade professional software, hoping to get your understanding and support.

Point 3: How you determine the concentration of zirconium ions in figure 4, please explain the accuracy of the experiments. By XRD you probably can determine the concentration of Cs2ZrCl6, the trend is similar that the one observed by ICP-AES. Please revise same issue in Figure 5.

Response 3: Thanks for your comments. Through XRD, the material composition analysis can be realized, and the form of zirconium ions can be known. Combined with ICP, the comprehensive quantitative analysis of zirconium ion concentration can be carried out.

Point 4: Please explain the selection of the parameters in the  electrolytic refining experiment.

Response 4: Thanks for your comments. I have systematically studied the various parameters of electrolytic refining experiments, and zirconium dendrites were obtained under different parameters. In this paper, only one parameter was selected for electrolytic refining. The purpose is to illustrate that the prepared composite electrolyte is suitable for electrolytic refining.

Point 5: Correct the numeration of figures, figure 8 and 6.

Response 5: Thanks for your comments. I have corrected it.

Point 6: Please explain the advantages of obtained zirconium dendrites and tehri role I the electrolyte.

Response 6: Thanks for your comments. The advantage of zirconium dendrites is that compared with powder zirconium products, it can avoid the problem of high oxygen content in the subsequent washing process. The advantages of the prepared electrolyte are strong high temperature stability and high zirconium ion concentration, which can effectively avoid the problem of large loss of ZrCl4 in the electrolysis process.

(2)    Conclusions

Point 1: Please explain the novelty of the review It is not clear which dendrites sizes are required and have already used.

Response 1: Thanks for your comments. Studies have shown that dendritic zirconium products can avoid the problem of high oxygen content caused by powder zirconium products in the subsequent washing process, so we want to obtain the largest dendritic size possible to reduce the oxygen content. At present, the industrial production in the field of zirconium electrolytic refining is mainly powdered zirconium products. It is of great significance to promote technological upgrading.

Point 2: Please revise the statement the maximum grain 264 size could reach 2.3 mm. It is not possible to observe grain in the SEM that you presented. Take in account the concept of particle size, grain size and crystallite size.

Response 2: Thanks for your comments. I have revised the statement and updated optical microscope images and SEM images. I seriously considered the concept of particle size, grain size and microcrystalline size, and modified it to dendrite size.

Reviewer 2 Report

In the given work the authors have prepared NaCl-KCl-CsCl-Cs2ZrCl6 com-
posite electrolyte, characterized with XRD and SEM etc. for possible applications. It is difficult for me to consider the given manuscript for publication because it lacks novelty, suitable literature review, required characterizations. However, it can be reconsidered if following points are thorougly addressed.

1. Introduction section requires  modern research work related to the given compositions and topic.

2. Novelty of the proposed work is not clear.

3. XRD for different composition has been done and just phase is discussed. Why other parameters like crystallite size, density have been ignored when you are only characterizing the material through XRD.

4. The authors have just provided the SEM and EDS of only a single sample which is not enough.

5. DTA/TGA analysis should be provided to get further insights.

6. The authors should make significant improvements in results & discussion and highlight them in revised manuscript.

Author Response

Response to Reviewer 2 Comments

Dear reviewer:

Re: Manuscript ID: materials-2216358 and Title: Preparation and application of NaCl-KCl-CsCl-Cs2ZrCl6 composite electrolyte.

Thank you from the bottom of my heart for your recognition and comments on our manuscript. Those comments are valuable and helpful. We have read through comments carefully and have made corrections. Based on your valuable comments, we have uploaded the revised manuscript file. Revisions in the text are shown using red highlight for additions, and strikethrough font for deletions. The responses to the your comments are marked in red and presented following.

We highly appreciate your time and consideration.

Sincerely.

WenZhen Zou.

Point 1: Introduction section requires modern research work related to the given compositions and topic.

Response 1: Thanks for your comments. I have given some modern research work related to the given compositions and topic in the 82-89 line of the article.

Point 2: Novelty of the proposed work is not clear.

Response 2: Thanks for your comments. I reiterated the novelty of the proposed work in the 99-103 line of the article.

Point 3: XRD for different composition has been done and just phase is discussed. Why other parameters like crystallite size, density have been ignored when you are only characterizing the material through XRD.

Response 3: Thanks for your comments. Because my purpose is to obtain a composite electrolyte with high zirconium ion concentration and high temperature stability. Firstly, the material composition was analyzed by XRD to determine whether Cs2ZrCl6 with stronger stability was generated, and then the concentration of zirconium ions was quantitatively analyzed by ICP. Other parameters such as crystallite size and density have little to do with this study.

Point 4: The authors have just provided the SEM and EDS of only a single sample which is not enough.

Response 4: Thanks for your comments. I supplemented multiple samples, as shown in Figure 10.

Point 5: DTA/TGA analysis should be provided to get further insights.

Response 5: Thanks for your comments. The lowest eutectic point and other properties of NaCl-KCl-CsCl molten salt system were analyzed by TG-DSC. The results are shown in Fig. 1.

Figure 1. TG-DSC analysis of NaCl-KCl-CsCl molten salt under argon protection

The results of TGDSC analysis showed that the ternary molten salt system NaCl-KCl-CsCl containing CsCl had a minimum melting point of 481.36°C, and the working temperature range of the subsequent preparation of NaCl-KCl-CsCl-Cs2ZrCl6 composite electrolyte was determined to be 488~750°C. It provide guidance for the preparation and use of my subsequent composite electrolytes.

I am sorry that this part of the data was published in another forthcoming article, so it is not provided and cited in this article. When I wanted to further use the composite electrolyte prepared by TGDSC, I was told that due to the strong volatility of the chloride, the detection equipment had been contaminated and therefore no detection was provided. I sincerely hope to get your understanding and support.

Point 6: The authors should make significant improvements in results & discussion and highlight them in revised manuscript.

Response 6: Thanks for your comments. I have made some improvements in results & discussion, as shown in the red part of the article.

Reviewer 3 Report

The submitted manuscript entitled “Preparation and application of NaCl-KCl-CsCl-Cs2ZrCl6 composite electrolyte“ presents experimental investigation on synthesis of the given composite electrolyte that can be used for obtaining or refining of Zr. The paper mainly describes optimization of various experimental parameters (such as composition, reaction temp. and time) used for the preparation of the aforementioned electrolyte. The manuscript should be considered for publication in Materials after minor corrections.

1.English should be improved. Lines 13-16, 131, 146-148, 161-166, 179, 188 etc. are some of my notices that should be corrected. Also, please watch for numbering in Figure captions.

2.Most of the discussion covers phase analysis obtained after different conditions. In order to complete this analysis the authors are advised to perform quantitative determination of the obtained phases (in wt%). Semi-quantitative analysis is achievable by XRD and with ICP combined the authors are in position to perform full quantitative analysis.

Author Response

Response to Reviewer 3 Comments

Dear reviewer:

Re: Manuscript ID: materials-2216358 and Title: Preparation and application of NaCl-KCl-CsCl-Cs2ZrCl6 composite electrolyte.

Thank you from the bottom of my heart for your recognition and comments on our manuscript. Those comments are valuable and helpful. We have read through comments carefully and have made corrections. Based on your valuable comments, we have uploaded the revised manuscript file. Revisions in the text are shown using red highlight for additions, and strikethrough font for deletions. The responses to the your comments are marked in red and presented following.

We highly appreciate your time and consideration.

Sincerely.

WenZhen Zou.

Point 1: English should be improved. Lines 13-16, 131, 146-148, 161-166, 179, 188 etc. are some of my notices that should be corrected. Also, please watch for numbering in Figure captions.

Response 1: Thanks for your notices and I have corrected them. Also, I updated the number of the Figure.

Point 2: Most of the discussion covers phase analysis obtained after different conditions. In order to complete this analysis the authors are advised to perform quantitative determination of the obtained phases (in wt%). Semi-quantitative analysis is achievable by XRD and with ICP combined the authors are in position to perform full quantitative analysis.

Response 2: Thanks for your comments. The reason why XRD is not used for semi-quantitative analysis is that the results are not ideal, which may be related to the RIR value. So I conduct quantitative analysis through the following steps: firstly, the material composition was analyzed by XRD to determine the form of zirconium ions, and then the concentration of zirconium ions was quantitatively analyzed by ICP. I sincerely hope to get your understanding and support.

Reviewer 4 Report

The authors reported the composition for preparing a composite electrolyte for the electrolytic refining of zirconium. This work is worthy of publication after making some minor changes in the manuscript.

11. There are grammatical errors throughout the manuscript that need improvement.

22. There are some numbering issues in Figures. For example, Fig. 6 on page 7, and again Fig 6 on page 9. Similarly, on the line umber 239 it is mentioned Fig. 8, but there is no Fig. 8 in the manuscript.

33. Also Fig. 10 is mentioned on the line number 249. There is no figure named as Fig. 10.

    Therefore, the figure numbers must be corrected in the Figure captions as well as in the text of the manuscript.

Author Response

Response to Reviewer 4 Comments

Dear reviewer:

Re: Manuscript ID: materials-2216358 and Title: Preparation and application of NaCl-KCl-CsCl-Cs2ZrCl6 composite electrolyte.

Thank you from the bottom of my heart for your recognition and comments on our manuscript. Those comments are valuable and helpful. We have read through comments carefully and have made corrections. Based on your valuable comments, we have uploaded the revised manuscript file. Revisions in the text are shown using red highlight for additions, and strikethrough font for deletions. The responses to the your comments are marked in red and presented following.

We highly appreciate your time and consideration.

Sincerely.

WenZhen Zou.

Point 1: There are grammatical errors throughout the manuscript that need improvement.

Response 1: Thanks for your comments. I tried my best to correct the grammatical errors.

Point 2: There are some numbering issues in Figures. For example, Fig. 6 on page 7, and again Fig 6 on page 9. Similarly, on the line umber 239 it is mentioned Fig. 8, but there is no Fig. 8 in the manuscript.

Response 2: Thanks for your comments. According to your instructions, I updated the number of the figure.

Point 3: Also Fig. 10 is mentioned on the line number 249. There is no figure named as Fig. 10.    Therefore, the figure numbers must be corrected in the Figure captions as well as in the text of the manuscript.

Response 3: Thanks for your comments. According to your instructions, I updated the number of the figure.

Round 2

Reviewer 2 Report

I am satisfied with the corrections made by the authors. The article can be accepted for publication.